

# Muscle strength and inflammatory response to the training load in rowers

Tomasz Podgórski[1], Alicja Nowak[2], Katarzyna Domaszewska[1], Jacek Mączyński[3], Magdalena Jabłońska[3], Jarosław Janowski[4] and Małgorzata B. Ogurkowska[3]

[1] Department of Physiology and Biochemistry, Poznan University of Physical Education, Poznań, Poland
[2] Department of Hygiene, Poznan University of Physical Education, Poznań, Poland
[3] Department of Biomechanics, Poznan University of Physical Education, Poznań, Poland
[4] Department of Theory of Sport, Poznan University of Physical Education, Poznań, Poland

## ABSTRACT

**Background**. Regular exercise leads to changes in muscle metabolism. The consequence of this is the adaptation to higher training loads. The aim of this study was to evaluate biomechanical and biochemical parameters describing the functions of skeletal muscles in periods when changes in training forms were introduced.

**Methods**. Seventeen male sweep-oar rowers, members of the Polish national rowing team, participated. The study was carried out at the beginning and at the end of the preparatory period. In the first and second examination measurements of torques of selected muscle groups and blood biochemical analysis were performed.

**Results**. There was observed a statistically significant decrease in the relative global force of the right lower limb between both terms of examination. A statistically significant increase in maximum torque was found for torso flexors. In the case of muscles responsible for torso rotation, a statistically significant decrease in the torque values of right torso rotators was observed. A significant difference was found with respect to creatine kinase activity, total testosterone concentration, total testosterone to cortisol ratio and total phenolics concentration ($p < 0.05$).

**Conclusion**. The study shows that the rowers' training should be more focused on building the strength of lower limbs to prevent the overload of lumbar spine and that the amount of force developed may be significantly affected by the antioxidant potential of rowers.

Corresponding author
Tomasz Podgórski,
podgorski@awf.poznan.pl

# INTRODUCTION

Rowing training is a significant load on the organism due to a high volume and intensity of training sessions, especially during the preparatory period. Studies conducted by some authors among well-trained rowers showed an increase in fatigue in certain training periods (*Woods et al., 2017*) and disruption to muscle function (*Gee et al., 2016*). This may be due to the fact that high-performance rowers engage about 70% of all the muscles when rowing a boat (*Richter, Hamilton & Roemer, 2010*).

Correctly planned sports training sessions result in adaptive changes in muscle tissue, which consist among other things in the improvement of metabolic processes and changes

in muscle structure. These changes cause an increase in muscle strength and/or increased resistance of muscle tissue to fatigue (*Egan & Zierath, 2013*). This process involves, among others, oxygen free radicals (reactive oxygen species, ROS) generated during physical activity (*Steinbacher & Eckl, 2015*), growth factors and inflammatory agents resulting from muscle fibre damage. (*Sass et al., 2018*; *Yang & Hu, 2018*). The number of granulocytes, monocytes and lymphocytes in the circulatory system increases and so does the concentration of some inflammatory factors e.g., interleukin 6 (IL-6) and acute phase proteins such as C-reactive protein (CRP) (*Pedersen & Hoffman-Goetz, 2000*; *Sass et al., 2018*).

Growth factors, free radicals and cytokines produced in damaged skeletal muscles initiate repair processes. However, with very intensive workouts, there may be an imbalance between these processes (*Steinbacher & Eckl, 2015*; *Tidball, 2005*). Systemic factors which are modified by physical activity, such as some hormones (testosterone, cortisol), endogenous and exogenous antioxidants or metabolic factors involved in myocyte energy metabolism, also play an important role in the mechanism regulating the function and structure of skeletal muscles (*Jensen et al., 2011*; *Pedersen & Hoffman-Goetz, 2000*; *Steinbacher & Eckl, 2015*).

Testosterone is the basic hormone involved in maintaining skeletal muscle structure and the related muscle strength. In the cardiovascular system there is mainly testosterone bound to specific proteins (SHBG, sex hormone binding globulin) and free testosterone. Both of these forms take part in the synthesis of muscle proteins (*Griggs et al., 1989*). Furthermore, testosterone activates the glucose metabolism-related signalling pathway in skeletal muscle cells via regulation of glucose transporter-4 (GLUT-4), which plays an important role in supplying skeletal muscles with energy substrates when the muscles are working (*Sato et al., 2008*).

Observation of parameters characterizing the functional state of skeletal muscles, including the strength of selected muscle groups or particular biochemical indices, allows to evaluate the athletes' degree of training adaptation. Skeletal muscle strength is most frequently assessed in isometric tests (*Hughes et al., 1999*; *Janiak, Wit & Stupnicki, 1993*; *Sterkowicz et al., 2018*) and isokinetic tests (*Buśko et al., 2018*; *Calmels et al., 1997*), depending on the study objective. Isometric and isokinetic muscle diagnostic tests were performed for the purposes of the study of overloading of the spine and joints caused by muscle strength deficits (*Diamond et al., 2016*; *Kocjan & Sarabon, 2014*; *McGregor, Hill & Grewar, 2004*). Comparative analysis of muscle strength topography for the studied groups allows to define the risk of sustaining musculoskeletal injuries (*Diamond et al., 2016*; *McGregor, Hill & Grewar, 2004*).

Due to the occurrence of frequent musculoskeletal injuries among professional rowers, in particular affecting the spine, the aim of this study was to evaluate biomechanical and biochemical parameters describing the functions of skeletal muscles in periods when changes in training forms were introduced. Discovering the pathobiomechanism of overload changes of the motor system in this group of athletes may allow to modify the training scheme, thereby leading to a reduction in injuries.

## MATERIALS & METHODS

### Participants

Initially, the study included 23 male sweep-oar rowers. A total of 17 subjects qualified for the study after excluding athletes who were injured, those who were taking anti-inflammatory drugs and those who did not fulfill the conditions of the study protocol (e.g., non fasting). The studied athletes were members of the Polish national rowing team who participate in major international competitions (World Championships, European Championships, Olympic Games). The study were carried out in two terms of the rowing training cycle: at the beginning (November) and at the end (March/April) of the preparatory period. The study was approved by the Regional Bioethics Committee at Poznan University of Medical Sciences (No. 208/17). Each of the athletes, having familiarized themselves with the study protocol, agreed in writing to participate in the study.

### Anthropometrics measurements

Body composition was measured in the first and second examination, with the use of the Tanita BC-418 MA analyser (Japan) with the subjects barefoot and wearing their underclothes. Body height was measured with the use of the WPT 60/150 OW medical scales (Radwag®, Poland).

### Biomechanics measurements

Biomechanical measurements of selected muscle groups were performed with the use of two devices; the first was produced by JBA Zb. Staniak (Poland) and was equipped with torque gauges with a relative measurement error of less than 0.5% (Figs. 1 and 2) and the second served to measure the global extension force of the lower limbs (Fig. 3). This device was equipped with a Scaime SB30X dynamometer (relative force measurement error of ± 0.017%) and a PUE 1 gauge (Radwag®, Poland).

The global force of lower limbs extensors is the resultant force generated simultaneously by extensors of hip, knee and ankle joints. Lower limb were measured asymmetrically, i.e., alternately left/right limb, respectively.

The construction of the devices ensures optimal stabilization of the limbs and torso and the possibility of taking measurements in body positions which are normally subject to measurement in biomechanical laboratories. The global force of lower limb extensors (*Kabacinski et al., 2018*) and torque of flexors, extensors and torso rotators under static conditions triggered in 1.5–3.0 s. were recorded. Each type of measurement was repeated 3 times. The final result was the highest of the 3 recorded values of force or torque, normalized to body weight.

### Blood collection and biochemical measurements

The athletes did not train on the day preceding the blood test. Blood samples were taken from the ulnar vein using a S-Monovette syringe tube (SARSTEDT, Germany), then placed in tubes containing a clot activtor and K3EDTA as anticoagulant, then centrifuged (1,500

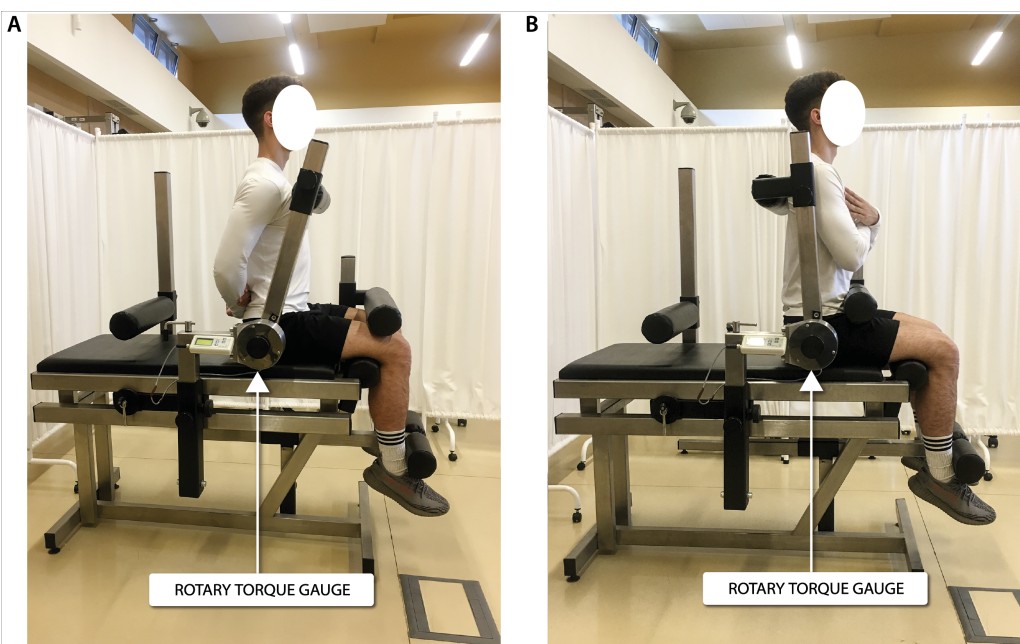

**Figure 1** Torque of torso flexors (A) and extensors (B) measurement in a seated position.

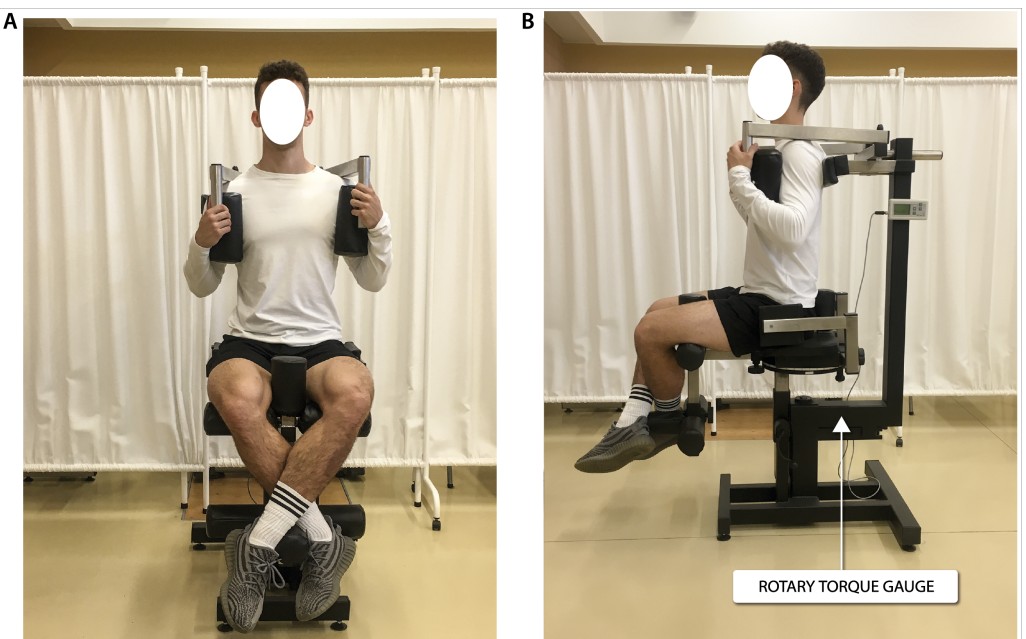

**Figure 2** Torque of torso rotators muscles measurement in a seated position in (A) frontal view and (B) sagittal view.

g, 4 °C, 4 min) in order to separate the serum and plasma, respectively. The samples were frozen and stored at −75 °C until the time the analyses were performed.

Creatine kinase (CK) activity and glucose concentration were determined with the use of the Accent 220S (Cormay, Poland) biochemical analyser and sets of enzymatic reagents

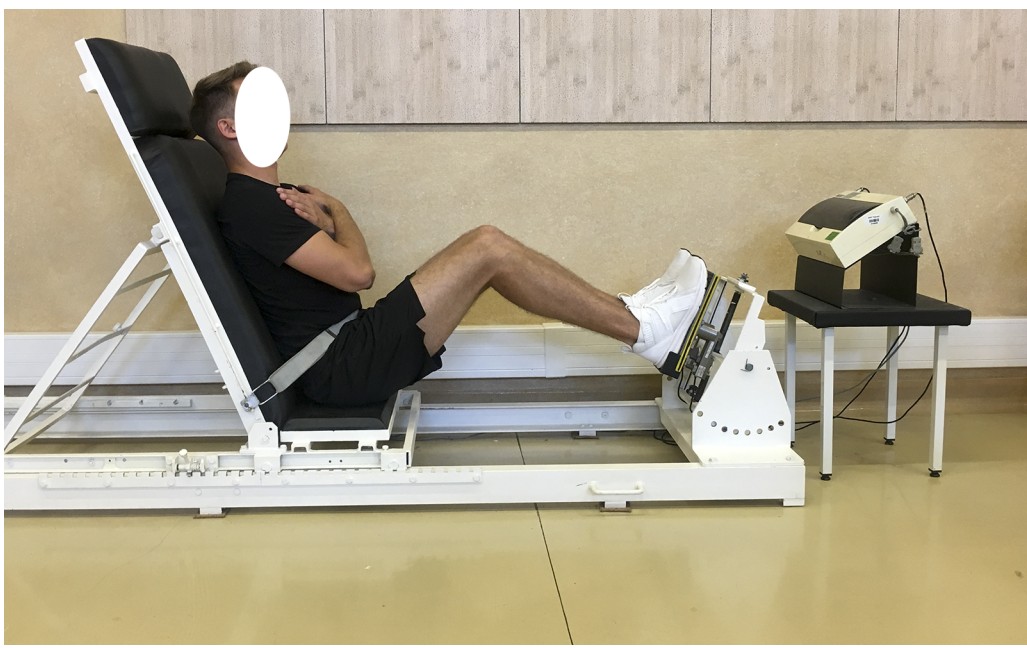

**Figure 3** Global force of lower limb extensors measurement in a sitting position.

(Cormay, Poland); the sensitivity of the sets was 7.4 U/L and 2.2 mg/dL, respectively. The concentrations of total testosterone (TT) and free testosterone (FT), cortisol (C), high-sensitivity C-reactive protein (hsCRP), insulin (DRG Instruments GmbH, Germany) and interleukin 6 (IL-6, AssayPro LLC, St. Charles, MO) were measured with the use of the ELISA immunoenzymatic method. The sensitivity of the ELISA kits was as follows: 0.083 ng/mL, 0.04 pg/mL, 2.5 ng/mL, 0.1 mg/L, 1.76 µIU/mL, 0.007 ng/mL, respectively.

Insulin sensitivity was expressed as a QUICKI (Quantitative Insulin Sensitivity Check Index) value and was calculated using the following formula: 1/log insulin + log glycemia in mg/dL (*Katz et al., 2000*).

Colometric methods were used to determine the concentrations of the total antioxidant capacity of plasma (Ferric Reducing Ability of Plasma, FRAP, reference values: 600–1600 µmol/L) (*Benzie & Strain, 1996*), the plasma concentration of thiobarbituric acid reactive substances (TBARS, reference values: 1–6 µmol/L) (*Ohkawa, Ohishi & Yagi, 1979*) and total phenolics (reference values: 2.8–4.0 g/L) (*Singleton & Rossi, 1965*).

## Training program

The training loads during the preparatory period are listed in Table 1. The loads were highly varied in terms of their direction and training modalities applied. The standard number of training sessions was from 9 to 10 during the preparatory period and from 12 to 14 during training camps. At the beginning of the preparatory period, general training sessions were prevailing with training modalities such as: running, proprioceptive exercises, swimming, cycling, cross-country skiing, team sports and strength training. Targeted training, i.e., rowing on a indoor rower, was included beginning with microcycle 5. Specialised exercises

Podgórski et al. (2020), *PeerJ*, DOI 10.7717/peerj.10355

Peer⌡

**Table 1 Rowers' weekly training frequency in a macrocycle (training load description).**

| Training objective | Week | Method | Training modality | Duration (Min.) | Intensity (%Max) | No. of training sessions per week |
|---|---|---|---|---|---|---|
| Aerobic endurance (maintenance) | 1–4 | constant | running, indoor rower, swimming | 100–125 | 60 | 5 |
| | 5–8 | constant | running, indoor rower | 100–125 | 60 | 3 |
| | 9–12 | constant | indoor rower, running | 80–120* | 60–65 | 3 |
| | 13–16 | constant/variable | indoor rower | 80–90 | 60–65 | 3 |
| | 17–20 | constant/variable | running, indoor rower, boat rowing | 60–90 | 60–65 | 3–4 |
| | 21–24 | constant/variable | boat rowing/running | 60–80 | 60–65 | 2 |
| Aerobic endurance (development) | 5–8 | constant/variable | running, bike, swimming, indoor rower, crosscountry skiing | 100–120 | 65–70 | 2 |
| | 9–12 | constant/variable | running, indoor rower | 90–110 | 65–70 | 2 |
| | 13–16 | constant/variable | running, indoor rower | 90–110 | 65–75 | 2 |
| | 17–20 | constant/variable | indoor rower/boat rowing | 90–100 | 65–75 | 1–2 |
| | 21–24 | constant/variable | boat rowing/running | 60 | 65–75 | 1 |
| Anaerobic endurance | 21–24 | short interval | boat rowing | 20–30 | 75–90 | 2 |
| Muscle mass -hypertrophy | 1–4 | body building | strength exercises | 120 | 60–80 | 3 |
| Maximum strength | 5–8 | strength athletics | strength exercises | 120 | 95–100 | 3 |
| Strength endurance | 13–16 | circuit training | strength exercises | 90–110 | 50–60 | 2 |
| | 17–20 | circuit training | | 90–110 | 50–60 | 2 |
| | 21–24 | circuit training | | 45–60 | 50–60 | 2 |
| Power | 9–12 | strength-speed | strength exercises | 100 | 60–70 | 3 |
| | 13–20 | | | 100 | 60–70 | 1 |
| | 21–24 | | | 45 | 60–70 | 1 |
| Deep muscle training | 1–12 | repetition | exercises with various beams, balance exercises, etc. | 30 | – | 1–2 |
| Team sports | 1–20 | variable | football, volleyball | 30–60 | 60–70 | 2 |

**Notes.**

Intensity % $VO_2$ max, strength training: % 1RM.

**Table 2  Somatic indices during first and second examination in groups of rowers ($n = 17$), mean (SD); median (interquartile range).**

| Parameters | Assessment First examination | Assessment Second examination |
|---|---|---|
| Age (yrs) | 22.23(4.58); 20.0(20.0–23.0) | |
| Training experience (yrs) | 9.35(4.45); 8.0(7.00–11.0) | |
| Body height (m) | 1.93(0.06); 1.92(1.89–1.94) | |
| Body mass (kg) | 93.65(8.80); 92.1(88.5–98.0) | 93.49(7.78); 92.0(90.5–98.4) |
| Fat mass (%) | 13.28(3.67); 11.4(10.4–16.0) | 12.57(3.47); 12.2(11.4–13.9) |
| Fat mass (kg) | 12.50(4.01); 10.5(10.0–14.0) | 11.74(3.48); 11.0(10.7–12.8) |
| Fat free mass (kg) | 81.15(7.71); 81.6(78.5–84.6) | 81.77(7.64); 81.8(78.3–84.8) |

on water were included from microcycle 17 and the amount of such training increased dynamically in subsequent microcycles.

At the beginning of the preparatory period (microcycles 1 to 4), there were mainly low and moderate intensity aerobic exercises and muscle mass building strength training. In the next mesocycle (microcycles 5 to 8) the intensity of exercises increased, the aim being to develop aerobic endurance, aerobic and anaerobic endurance and maximum strength. This trend continued in the next mesocycle (microcycles 9 to 12), however, with a significant increase in aerobic-anaerobic loads (2 sessions in the microcycle). Power developing sessions were also introduced. In microcycles 13 to 24 the development of aerobic and aerobic-anaerobic endurance continued and sessions increasing anaerobic endurance were introduced. In strength training, power training continued and strength conditioning components were included. The detailed training programme is presented in Table 1.

## Statistical analysis

All data are presented as mean, standard deviation (SD) and median values and interquartile range. The normality of distribution was tested with the *Shapiro–Wilk* test. The differences between paired and normally distributed variables were investigated using the *T*-test and the *Wilcoxon* test was used for non-normally distributed variables. The *Pearson* analysis for normally distributed variables and *Spearman's rank* analysis for non-normally distributed variables were used to calculate correlation coefficients. The level of statistical significance was set at $p < 0.05$. The obtained results were analysed statistically using the Dell Inc. (2016) Dell Statistica 13 software (Tulsa, Oklahoma, USA).

## RESULTS

There was no significant difference between the first and second examination with respect to the somatic features of the rowers (Table 2).

With respect to biomechanical indices (Table 3), a statistically significant decrease in the relative global force of the right lower limb from the beginning to the end of the preparatory period was observed (mean value decreased by 2.7 N/kg). In the case of the lower left limb there was a tendency for similar changes ($p = 0.09$) (mean value decreased by 2.0 N/kg).

**Table 3  Biomechanical indices during first and second examination in groups of rowers ($n = 17$), mean (SD); median (interquartile range).**

| Parameters | Assessment First examination | Assessment Second examination |
|---|---|---|
| Global strength of lower limb extensors, left (N/kg) | 31.8 (6.8); 29.7 (25.5–37.1) | 29.8 (5.1); 29.4 (25.5–34.4) |
| Global strength of lower limb extensors, right (N/kg) | 31.3 (5.3); 30.8 (27.5–34.6) | 28.6 (3.5); 28.3 (25.9–31.4)[*,a] |
| Torques - torso, extensors (Nm/kg) | 7.06 (1.30); 6.89 (6.25–7.93) | 7.45 (1.09); 7.31 (6.80–7.70) |
| Torques - torso, flexors (Nm/kg) | 3.44 (0.31); 3.41 (3.20–3.54) | 3.80 (0.33); 3.69 (3.56–3.97)[*,b] |
| Torques - torso rotators, left (Nm/kg) | 1.46 (0.25); 1.46 (1.31–1.71) | 1.39 (0.31); 1.42 (1.20–1.61) |
| Torques - torso rotators, right (Nm/kg) | 1.33 (0.24); 1.32 (1.16–1.40) | 1.24 (0.26); 1.18 (1.10–1.33)[*,a] |

Notes.
[*]significance of the difference between the first and second examination ($p < 0.05$).
[a]*T*-test.
[b]*Wilcoxon* test.

However, taking into account the group of torso muscles, a statistically significant increase in maximum torque between the first and second examination (Table 3) was found for torso flexors (mean value increase by 0.36 Nm/kg). In the case of extensors there was a tendency towards change ($p = 0.09$) (increase by 0.39 Nm/kg). In the case of muscles responsible for torso rotation, a statistically significant decrease in the torque values of right torso rotators was observed between the two examinations (decrease in mean value by 0.09 Nm/kg). There were no significant changes between the study of other biomechanical parameters.

A comparative analysis of the biochemical indices (Table 4) measured showed a significant difference with respect to CK, TT, TT/C and total phenolics ($p < 0.05$).

In the first examination, the correlation analysis showed relationships between the concentration of total phenolics and the global strength of extensors of the left lower limb ($r = 0.54$, $p = 0.026$) and the right lower limb ($r = 0.61$, $p = 0.009$, Fig. 4A) and the torque of right torso rotators ($r = 0.51$, $p = 0.035$, Fig. 4B). The fat mass correlated with the QUICKI index ($r = 0.55$, $p = 0.023$).

Changes ($\Delta_{II-I}$) in TT concentration between the two examinations correlated with insulin ($r = -0.54$, $p = 0.026$) and glucose ($r = -0.52$, $p = 0.032$) concentrations in the second examination, and the change ($\Delta_{II-I}$) in the trunk extensors torque correlated with the TT/C ratio ($r = -0.50$, $p = 0.042$) and FT/C ($r = -0.50$, $p = 0.042$) in the second examination (Fig. 4C).

## DISCUSSION

This study presents the effect of intense rowing training in the preparatory period on selected indicators of muscle damage and muscle tissue repair mechanisms. A specific change in biomechanical parameters describing the strength of selected groups of muscles of the torso and lower extremities was observed. A decrease in the relative global force of extensors of the lower extremities was observed with a significant change in the right extremity. The mean global force of the extensors decreased by 9.7% for the right lower limb and by 7.1% for the left lower limb. In terms of the torso, there was a significant increase in the torque of flexor muscles (9.3% on average) and a tendency towards changes

**Table 4** Biochemical indices during first and second examination in groups of rowers ($n = 17$), mean (SD); median (interquartile range).

| Parameters | Assessment First examination | Assessment Second examination |
| --- | --- | --- |
| hsCRP (mg/L) | 0.19(0.07); 0.18(0.15–0.22) | 0.16(0.09); 0.15(0.07–0.23) |
| IL-6 (ng/mL) | 0.09(0.05); 0.07(0.07–0.10) | 0.09(0.05); 0.07(0.06-0.11) |
| Insulin (μIU/mL) | 10.34(2.37); 9.81(8.83–10.93) | 9.89(2.29); 9.36(8.01–10.70) |
| Glucose (mg/dL) | 96.18(6.73); 97.0(90.0–100.0) | 95.29(8.35); 96.0(93.0–102.0) |
| QUICKI | 0.34(0.01); 0.34(0.33–0.34) | 0.34(0.01); 0.34(0.33–0.34) |
| CK (U/L) | 437.1(313.9); 313.1(209.7–480.5) | 266.1(174.1); 208.1(154.9–275.8)[*,b] |
| C (ng/mL) | 315.5(191.3); 253.4(235.1–294.8) | 292.7(90.9); 274.5(229.6–331.8) |
| TT (ng/mL) | 5.74(1.13); 5.54(4.97–6.29) | 5.21(1.27); 5.17(4.45–5.59)[*,b] |
| FT (pg/mL) | 16.31(4.06); 15.88(13.36–17.95) | 16.43(4.54); 15.55(13.33–18.94) |
| TT/C | 0.023(0.010); 0.02(0.018–0.025) | 0.020(0.008); 0.02(0.014–0.023)[*,a] |
| FT/C | 0.065(0.031); 0.07(0.053–0.074) | 0.061(0.025); 0.06(0.045–0.069) |
| FRAP (μmol/L) | 861.0(146.6); 828.5(762.4–938.3) | 855.6(160.2); 872.0(764.6-938.6) |
| Total phenolics (g GAE/L) | 2.51(0.10); 2.50(2.42–2.57) | 2.59(0.16); 2.61(2.54–2.64)[*,a] |
| TBARS (μmol/L) | 4.54(1.12); 4.16(3.93–5.17) | 4.10(0.97); 3.87(3.38–4.45) |

**Notes.**

Abbreviations: hsCRP, high sensitivity C-reactive protein; IL-6, human interleukin-6; QUICKI, Quantitative Insulin Sensitivity Check Index; CK, creatine kinase; C, cortisol; TT, total testosterone; FT, free testosterone; FRAP, Ferric Reducing Ability of Plasma; GAE, gallic acid equivalent; TBARS, thiobarbituric acid reactive substances.

[*]indicates a significant difference ($p < 0.05$).

[a]*T*-test.

[b]*Wilcoxon* test.

in rectifiers (by 4.9%). In the analysed period of the training cycle the opposite effect should be expected. Namely, at the end of the preparatory period, lower limb extensor muscles, which are most involved in the pulling phase in the rowing cycle should definitely become stronger compared to the beginning of the preparatory period. The results are probably related to the fact that at the beginning of the preparatory period (weeks 1 to 4) the athletes underwent strength training with a particular emphasis on the mass and strength of the lower limbs (see training description, Table 1). This kind of training was absent in the subsequent stages of the preparatory period and was replaced by exercises on indoor rowers and rowing the boat in natural conditions, i.e., on the water. This type of training entails a change in the load on the lower limbs and skeletal muscles of the torso are involved to a large extent; they become stiffened over time, which reduces their ability to reach maximum torques, i.e., they become overloaded. The above phenomenon concerns mainly extensor muscles which –in the case of inadequately trained lower limb muscles –are engaged to generate the force necessary to propel the boat (*Ogurkowska, Kawałek & Zygmańska, 2015*). The above fact confirms one of the findings of the study, i.e., that the increase in torso torque was much smaller in the case of extensor muscles than flexor muscles.

There was also a decrease in the values of torso torque between the first and second examination (mean values decreased by 7.5% for the right rotator and by 9.3% for the left rotator). The technique of rowing with sweep oars requires asymmetrical work, especially when the water is "catching" the blade, during which the body rotates towards the outrigger,

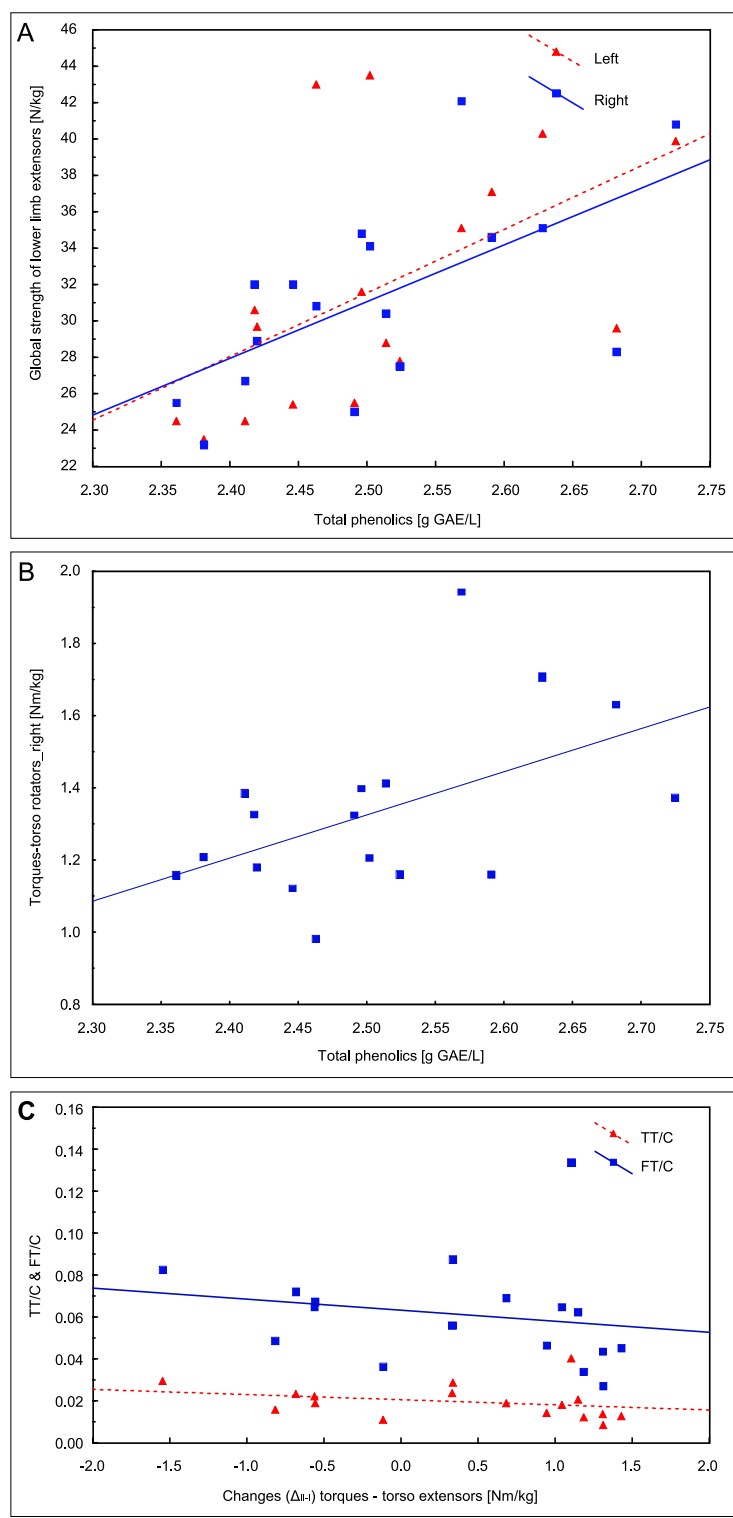

**Figure 4** *Pearson* correlations of the total phenolics concentration with the global strength of extensors of lower limbs (A) and the torque of right torso rotators in the first examination (B); (C) *Spearman's rank* correlation of the change between two examinations ($\Delta_{II-I}$) in the trunk extensors torque with total and free testosterone to cortisol ratios (TT/C and FT/C) in the second examination.

so that the shoulders are as parallel to the oar as possible. The inner shoulder is lowered due to the need to rotate the body, while the outer shoulder is slightly higher. This means that while rowing, the torso rotators, as well as flexors and extensors, are overloaded, even more so after prolonged training. It should be assumed that this is the case during the second half of the analysed period of the training cycle, which focuses primarily on the indoor rower and boat. It was noted that during a maximal rowing trial on the indoor rower, lumbar flexion may increase from 75% to 90% of the maximum range of motion, most likely due to muscle fatigue (*Ogurkowska, 2007*). The load on the rotator muscles leads to their stiffening and, at the same time, to a reduction in the values of torque they reach. It should be emphasized that the obtained results may be indicative of the pathobiomechanism of overload changes in the lumbar vertebral column in professional rowers (*Hosea et al., 2011*; *Ogurkowska, 2007*; *Ogurkowska, Lewandowski & Kawałek, 2016*).

In the second examination, we noted a decrease in the concentration of total testosterone and the TT/C ratio, recognized as an anabolic/catabolic index. Other authors attributed such changes to body fatigue resulting from significant training loads (*Ahtiainen et al., 2003*). No significant changes in cortisol concentration and pro-inflammatory factors were observed in the study, which indicates that the decrease in the TT/C ratio value was probably not the result of the athletes' fatigue. In a study carried out on football players, Michailidis explained that the absence of changes in cortisol concentration in the annual training cycle was due to an appropriate psycho-physical preparation of the athletes (*Michailidis, 2014*). It is not impossible that the decrease in testosterone concentration was related to seasonal changes. In observational studies, *Reinberg & Lagoguey (1978)* showed that serum testosterone concentration in the annual cycle peaks in November. Moreover, in our study, despite the decrease in testosterone concentration in the second examination, we recorded an inverse correlation between TT/C and FT/C ratios and a change in the torque of the trunk extensor, which indicates that the slight changes in the mentioned hormone did not prevent the increase in muscle strength of the torso.

The decrease in creatine kinase activity between the two examinations proves that a higher load on muscle tissue engaged in the exercises occurred on the first examination. Creatine kinase activity depends on the duration and type of physical activity (*Banfi et al., 2012*). In particular, eccentric exercises lead to an increase in CK activity (*Clarkson & Hubal, 2002*; *Nosaka & Clarkson, 1994*). On the first examination the training of rowers was characterized by a larger volume and a significant share of strength training as compared to the second examination (Table 1), which may have had an impact on the changes in CK activity. It is also worth noting that on the second examination the global strength of the lower limbs decreased, which may be indicative of a decrease in the rowers' muscle mass engaged in the exercise, and which may also have had an impact on the changes in CK activity. Training adaptation resulting from the training program lasting several months could be an additional factor contributing to the decreased activity of this enzyme on the second examination of the study.

Creatine kinase activity reflects the level of muscle fibre damage resulting from intense muscle work and the rate of clearance of this enzyme from the blood. Elevated creatine kinase activity was observed for a few days after physical activity of sufficiently high intensity

and duration. According to suggestions, the CK activity should preferably be measured 24 to 48 h following physical activity (*Banfi et al., 2012*). In this study, blood was collected two days after the end of the last standard training session.

Release of CK from muscle cells during exercise corresponds to the degree of permeability of cell membranes and their damage resulting, among other things, from an increase in the amount of free radicals formed during physical activity, leading to the peroxidation of cell membrane lipids (*Banfi et al., 2012*; *Mason et al., 2016*). In our study we determined TBARS concentration reflecting the level of cell membrane damage. However, we did not find any significant changes in this indicator between the two examinations. At the same time, we observed a significant increase in the concentration of total phenolics, which are involved in scavenging of free radicals. In the data from the first examination, we found significant relationships between the concentration of these substances and some parameters of muscle strength (global strength of the left and right lower limb extensors and torque of right torso rotators), which may be indicative of their protective effect on skeletal muscle fibres. These substances in the circulatory system are an indicator of the amount of consumed foods which are rich in phenolic compounds and their absorption from the gastrointestinal tract (*Scalbert & Williamson, 2000*). Additionally, phenolic compounds have been shown to reduce muscle damage caused by increased ROS levels during physical exercise and restitution (*Malaguti, Angeloni & Hrelia, 2013*). It should be noted that total plasma antioxidant capacity (FRAP) did not differ significantly between the two examinations, which means that the prooxidant-antioxidant balance was maintained despite significantly higher CK activity in the case of the first examination.

Reactive oxygen species produced during exercise may contribute to an intensified release of pro-inflammatory factors. In the study we measured levels of IL-6 and hsCRP. It is well known that cytokine levels may be induced by intense physical exercise from two to six days after exercise (*Kasapis & Thompson, 2005*; *Pedersen & Hoffman-Goetz, 2000*). An increase in IL-6 concentration during muscle work is a result of two mechanisms: muscle contraction and muscle fibre damage. The level of increase in IL-6 concentration after physical exercise depends on the type of exercise, intensity and duration as well as the mass of skeletal muscle engaged (*Kasapis & Thompson, 2005*). In our study, we found no significant changes in the indices on the second examination compared to the first examination, which means that the decrease in the strength of lower limb muscles on the second examination of the study was related to the training technique rather than overloading.

It has been shown that many immune cell functional responses may be modulated by carbohydrate status. Blood glucose is the fuel for cells of the immune system. The liver and muscle glycogen stores play the important role in minimising immunosuppression (*Close et al., 2005*). In our study, however, we did not report any relationships between the concentration of pro-inflammatory indices and carbohydrate metabolism indices (glucose, insulin or QUICKI). We also did not notice any significant changes between the examinations of the values of these indices, which may be the result of high training adaptation of the athletes in terms of carbohydrate metabolism already on the first date of the examination. Training adaptation consists in improving the transport of glucose in

the insulin-independent pathway (*Egan & Zierath, 2013*). This mechanism is influenced a.o. by testosterone, which activates the glucose metabolism-related signalling pathway in skeletal muscle cells via regulation of glucose transporter-4 (GLUT-4) (*Sato et al., 2008*). The decrease in TT level on the second examination correlated with the concentration of glucose and insulin, but it did not significantly affect QUICKI.

In the examinations, we determined the concentration of total phenolics which show antioxidant effects and are an indicator of intake of foods rich in these compounds. However, we did not include an analysis of the athletes' diet, which could document the influence of the consumed foods on the level of antioxidant potential and its importance in the training process.

## CONCLUSIONS

The observed changes in biomechanical parameters during the training period show that technique-oriented training should be more focused on strengthening the athletes' lower limbs in order to protect the lumbar spine from overloading. It was also found that the amount of force developed is significantly related to the antioxidant potential of the body and we conclude that it may play a significant role in muscle strength. However, confirmation of this conclusion requires a larger group of subjects and representatives of various sports disciplines. Further studies are needed to investigate if there is a causality effect.

## ACKNOWLEDGEMENTS

The authors would like to thank the athletes who took part in this study. The authors also would like to thank Ms. Magdalena Lewandowska for her assistance in the statistical analysis.

### Funding

This study was funded by the Polish Ministry of Science and Higher Education within the "Development of Academic Sport" program (Project No. N RSA4 06154). The funders had no role in study design, data collection and analysis, decision to publish, or preparation of the manuscript.

### Grant Disclosures

The following grant information was disclosed by the authors:
Polish Ministry of Science and Higher Education within the "Development of Academic Sport" program: N RSA4 06154.

### Competing Interests

The authors declare there are no competing interests.

## Author Contributions

- Tomasz Podgórski and Alicja Nowak performed the experiments, analyzed the data, prepared figures and/or tables, authored or reviewed drafts of the paper, and approved the final draft.
- Katarzyna Domaszewska analyzed the data, prepared figures and/or tables, authored or reviewed drafts of the paper, and approved the final draft.
- Jacek Mączyński and Jarosław Janowski performed the experiments, analyzed the data, prepared figures and/or tables, and approved the final draft.
- Magdalena Jabłońska performed the experiments, analyzed the data, authored or reviewed drafts of the paper, and approved the final draft.
- Małgorzata B. Ogurkowska conceived and designed the experiments, performed the experiments, analyzed the data, prepared figures and/or tables, authored or reviewed drafts of the paper, and approved the final draft.

## Human Ethics

The following information was supplied relating to ethical approvals (i.e., approving body and any reference numbers):

The study was approved by the Regional Bioethics Committee at Poznan University of Medical Sciences (208/17).

## Data Availability

The raw measurements are available in a Supplemental File.

## Supplemental Information

Supplemental information for this article can be found online at http://dx.doi.org/10.7717/peerj.10355#supplemental-information.

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
