# Peer review of "Muscle strength and inflammatory response to the training load in rowers"

_PeerJ, doi:10.7717/peerj.10355_

## Round 0.1 · original submission · Minor Revisions

Dear Dr. Podgórski,

After the review by two independent reviewers, they consider your text is almost ready to be accepted for publication in PeerJ. Still, some minor changes are necessary before the text is accepted. Please revise your manuscript and resubmit up to 25th October 2020.

Sincerely,
Daniel Silva

·

Basic reporting

This research article presents original data about assessing the biomechanical and biochemical profile changes in training forms were introduced. Therefore, the study counted with seventeen male sweep-oar rowers, members of the Polish national rowing team. The authors examined anthropometric, biomechanical, and biochemical factors related to skeletal-muscle function and performance at two moments: 1) at the beginning of the preparatory period; and 2) at the end of preparatory period. In summary, the training program of approximately five months was sufficient to lead a significant (P<0.05) decrease in some biomechanical factors: global strength of lower limb extensors (right), torques – (torso, flexors), and torque (torso rotator, right); and some biochemical factors: CK, TT, and TT/C. Nevertheless, total phenolics increases significantly after the intervention. For this reason, the authors stated that the amount of strength might be significantly affected by the antioxidant defense promoted by excessive muscle damage.

The manuscript is well written and can improve physiologists and exercise sciences/health sciences professionals' understanding.

The article included a sufficient introduction and background to demonstrate how the work fits into the broader field of knowledge. Moreover, the relevant prior literature was appropriately referenced. However, there are some aspects regarding the validity of the finding that should be clarified better.

1) The central point that needs more clarification is why the authors stated in the conclusion that the antioxidant significantly affects the amount of force? The reason I am asking is that there is no relation of causality between those conditions. The authors indeed bring a contextualization that points to this affirmation, however, the increase of total phenolics presented in this research does not allow the affirmation that antioxidant defense is playing a role on muscle strength, it can be "Randomness." Therefore, I suggest rewriting the sentence suggesting the antioxidant profile as a possible variable that might be playing a role in muscle strength. Moreover, further studies should also investigate if there is a causality effect.

Experimental design

The study design is appropriate to address the primary aim of the study.

Validity of the findings

The validity of the findings agrees with the experimental design. The only correction needed is at the conclusion (mentioned right above).

Additional comments

There are a few minor comments:

1) add the sample size at the caption of all tables;
2) State the statistical test performed to find the P-value at the table legends.
3) Why did the authors not present a table or a figure with the correlation? This illustration may facilitate the reader.

·

Basic reporting

English is not my primary language. However, I find that the text presented for evaluation is fully understandable and the information contained in the chapters describes well the problems and methods presented in the article.
The authors used sufficiently extensive literature on the impact of intense exercise on biomechanical and biochemical changes in the body of highly qualified athletes. There are many publications on this subject, while this publication refers to the most important items in the literature on rowers.
The structure of the paper corresponds to the general principles of scientific publications, i.e. division into sections, references, tables and figures.
Raw data for biomechanical measurements and biochemical analyzes were provided in a form that could be reproduced and recalculated.
The experimental group of players, measurement methods and biochemical analyzes as well as the results and their adequate statistical analysis were well selected in order to verify the research hypotheses.
Some of the cited references indicate a tendency to "strengthen" the scientific argument carried out by the authors of this article. I did not notice the glaring "clustering" of similar thematically similar publications.

Experimental design

The article clearly presents two aims: i) research (scientific) and ii) practical (training). The first was to obtain information on the relationship between training in the preparatory period and changes in strength in selected skeletal muscle groups and the biochemical basis of these changes. The practical goal is to link the training loads with the value of strength gains and overtraining rates in the preparation period.
The applied measurement methodology meets all the highest standards in biomechanical and biochemical research. Stands for measuring force in isometric conditions are widely used in sports research and the implemented metrological solutions are patented. The scheme of the experiment is very simple, it consists in verifying the effects "before-after". The factor that initiates changes in the body of the studied persons is the training load in the preparation period.
The consent for the experiment with the participation of humans was issued by the Regional Bioethics Committee at Poznań University of Medical Sciences.
Standard biochemical methods were used, citing the references in which they were described. Similarly with the methodology of measuring muscle strength. The description in the text is supplemented with a view of the subject on the stand, illustrating the construction of the stand, positions during measurement and the method of stabilizing the adjacent body segments. In the description under Figures 1-3, it would be advisable to add the information "under static condition".

Validity of the findings

The value added to literature is the association / correlation of representative biomechanical and biochemical parameters with the content of rowing training. It should be emphasized that they were rowers of the highest sports level, often the best at world. Inducing an increase in physical fitness features in such highly qualified athletes with long-term training experience is a great challenge for training methods. This article provides the content and forms of training during the 24 weeks of the preparatory period. There are few publications presenting so precisely the training methodology not only in rowing but also in other sports.
Raw data is available and can be further processed. The applied statistical analysis is simple and accurately selected to verify the purpose of the work. It would be advisable to give in tab. 3 and tab. 4 information on which test (t-Student or Wilcoxon) detect significant differences in the relevant parameters. An additional comment to the statistical analysis is provided in the attached file.
The work is completed with short and careful conclusions regarding recommendations for improving the training process for this group of athletes. The conclusions follow directly from the results of the experiment. The few correlations in this work seem to indicate that the antioxidant potential of the organism has an influence on the value of the skeletal muscle force. However, confirmation of this conclusion requires a larger group of voluntary subjects, representatives of various sports disciplines.

Additional comments

The experiment is very interesting and scientifically valuable and brings new information to sports practice. It is necessary to run and check the statistical analysis of the results again.


SELECTED PROBLEMS RELATED TO THE STATISTICAL ANALYSIS OF THE RESULTS.
Table 4 shows the biochemical parameters that changed after training in the preparation period. These include: CK (U / L), TT (ng / ML), TT / C, Total phenolics (g GAE / L). Using Statistica v.13.3, the analyze has been re-calculated.
Shapiro-Wilk test t-Student Wilcoxon test
Examination
First Second
CK(U/L) p=0,0181 p=0,0010 p=0,0245
TT (ng/ML p=0,9366 p=0,0430 (-) (-)
TT/C p=0,3382 p=0,2305 p=0,0062 p=0,0615
Total phenolics (g GAE/L) p=0,4632 p=0,4610 p=0,0123


1) According to the raw data provided by the authors, the mean and standard deviation values for TT_1 and TT_2 are respectively:
TT_1 = 5,74±1,13, while in tab.4 = 5,81±1,13)
TT_2 = 5,21±1,27, while in tab.4 = 5,28±1,27)
Therefore, it is not known for what values the difference between the tests for this parameter was calculated.
2) It has to be decided which of the t-Student test or Wilcoxon test is to be used to analyze the differences in the TT/C parameter. The basis is the assessment of the normality distribution of the differences between the paired values of the first and second measurements (delta). The Shapiro-Wilk test shows that the distribution of empirical data (delta) is not close to the theoretical one because W = 0.9156 (p = 0.1251). A borderline value of p = 0.3 is assumed to assess the compliance of these distributions. Thus, for the TT / C parameter, the Wilcoxon test to evaluate the differences is adequate. As shown in the table above, it cannot be concluded that there are differences in the "before-after" measurements of TT / C.

---

## Round 0.2 · accepted · Accept

Congratulations! Both reviewers considered that your manuscript is now fit for publication in PeerJ.

·

Basic reporting

No comment

Experimental design

No comment

Validity of the findings

No comment

Additional comments

The authors successfully addressed all the reviewers comments, increasing the manuscript quality.
Thank you.

·

Basic reporting

The manuscript is prepared in the form of good standard scientific articles. Literature selected adequately to the subject and appropriately used in the chapter "discussion". There are no objections to the structure of the tables, which well illustrate the measurement results and their analysis. After specifying the purpose of the work as part of the self-correction, the final conclusions are clear.

Experimental design

The problem undertaken at work corresponds to the profile of the journal. The work is well grounded in the research on the impact of maximum physical effort on the human body. The original research methodology and properly selected statistical analysis of the results should be emphasized.

Validity of the findings

The results of this work cover a certain gap between the basic physiological knowledge and the training theory in competitive sport. The conclusions enrich the basic knowledge on the border of physiology and biochemistry and constitute the premises for the methodology of sports training.

Additional comments

Interesting publication. A well-planned experiment. It is worth emphasizing that measurements were carried out on such a large group of competitors of the highest sports level.